# Analysis of Quality Indicators of the Pre-Analytical Phase on Blood Gas Analyzers, Point-Of-Care Analyzer in the Period of the COVID-19 Pandemic

**DOI:** 10.3390/diagnostics13061044

**Published:** 2023-03-09

**Authors:** Vincenzo Brescia, Lucia Varraso, Mariantonietta Antonucci, Roberto Lovero, Annalisa Schirinzi, Elisa Mascolo, Francesca Di Serio

**Affiliations:** Clinical Pathology Unit, AOU Policlinico Consorziale di Bari–Ospedale Giovanni XXIII, 70124 Bari, Italy

**Keywords:** point-of-care (POC), EGA, GEM 4000, quality indicators, non-conformities (NC)

## Abstract

Aim of the study: We evaluated and compared blood gas analysis (EGA) non-conformities (NC) considered operator-dependent performed in Point-Of-Care (POC) analyzer as quality indicators (IQ) of the pre-analytical phase. To this end, four different NC registered in the resuscitation departments of the Hospital Polyclinic Bari from the beginning of the pandemic (March 2020) until February 2022 were evaluated. The results obtained were compared with those recorded in the pre-COVID period (March 2018–February 2020) to check if there were differences in number and type. Material and methods: GEM 4000 series blood gas analyzers (Instrumentation Laboratory, Bedford, MA, United States) are installed with integrated Intelligent Quality Management (iQM^®^), which automatically identify and log pre-analytical errors. All blood gas analyzers are connected to the company intranet and interfaced with the GEM Web Plus (Werfen Instrumentation Laboratory, Bedford, MA, United States) data management information system, which allows the core laboratory to remotely supervise all decentralized POC stations. The operator-dependent process NC were expressed in terms of absolute and relative proportions (percentiles and percentage changes). For performance evaluation, the Mann–Whitney U test, Chi-squared test and Six-Sigma Metric calculation for performance classification were performed. Results: In the COVID period, 31,364 blood gas tests were performed vs. 16,632 tests in the pre-COVID period. The NC related to the suitability of the EGA sample and manageable by the operators were totals of 652 (3.9%) and 749 (2.4%), respectively, in the pre-COVID and COVID periods. The pre-analytical phase IQs used did not show statistically significant differences in the two periods evaluated. The Sigma evaluation did not show an increase in error rates. Conclusions: Considering the increase in the number of EGAs performed in the two periods, the training procedures performed by the core laboratory staff were effective; the clinical users of the POC complied with the indications and procedures shared with the core laboratory without increasing the operator-dependent NCs. Furthermore, the core laboratory developed monitoring activities capable of guaranteeing the maintenance of the pre-analytical quality.

## 1. Introduction

The need to provide laboratory tests outside the core laboratory has led to the implementation of Point-Of-Care (POC) analyzer decentralized laboratory analyses. Point-of-Care (POC) analyzer is a proven approach able to provide faster turnaround times (TAT) of laboratory test results [1]. One of the main areas of application of decentralized analyzes in the hospital setting is represented by “Critical Emergency Medicine”, a medical specialty aimed at formulating a diagnosis of pathology quickly, followed by an equally timely therapeutic treatment to restore the state of health of the patient, maintain vital signs and prevent complications [2]. Blood gas analyzers (POC) [3] play a crucial role in the clinical diagnosis as well as appropriate therapeutic monitoring of critically ill patients with severe SARS-CoV2 infection admitted to intensive care units [4,5,6,7]. In addition to CLSI documents [8,9,10], other practical summaries [1,11] have been produced on the activities required to achieve high quality POC results that are accurate, precise and clinically valuable.

The aspects of quality assurance addressed in these documents include: (1) device selection, (2) device verification, (3) quality assurance including reagent and quality control (QC), lot changes and (4) quality management including the selection of appropriate Quality Indicators (QI) and operator and document management. The available evidence strongly suggests that the pre- and post-analytic are more error-prone steps than the intra-analytical phase in Total-Process Testing (TTP) [11,12]. As with other in vitro diagnostic tests, EGA can be prone to errors throughout the TTP, but in particular to incorrect management of the pre-analytical phase [13]. Furthermore, the guidelines highlight that the clinical users’ competence is an important factor influencing the qualitative level of the POC analytical performance [14,15,16].

The aim of this study was to evaluate and compare the non-conformities (NC) recorded by blood gas analyzers (EGA) performed in Point-Of-Care (POC) analyzer, considered operator-dependent. We verified whether these NC, the quality indicators (QI) of the pre-analytical phase, could be used to evaluate the quality level of performance and related clinical risk in a particular period of clinical activity such as the COVID-19 pandemic.

## 2. Materials and Methods

### 2.1. Patient Samples

The blood gas analysis results in the resuscitation departments of the General Hospital from the beginning of the COVID-19 pandemic (March 2020) until February 2022 were collected. The results obtained were then compared with those recorded in the pre-pandemic COVID-19 period (March 2018–February 2020). The study was approved by the ethics committee of Bari (Italy) (N. 6388 COVID19 DOM protocol number 0034687/12-05-2020) and written informed consent was obtained.

The results of the individual blood gas analyses were evaluated to verify the presence of operator-dependent process non-conformities.

### 2.2. Analytical Systems

The Resuscitation departments of the Hospital Policlinic of Bari Italy are equipped with three blood gas analyzers of the GEM 4000 series (Werfen Instrumentation Laboratory, Bedford, MA, USA). The blood gas analyzers of the GEM 4000 series, used in the study, are in vitro diagnostic systems with innovative technology that allows blood gas analysis to be performed on minimal volumes (about 150 µL) of heparinized whole blood samples, providing results quickly (within 1 min). The GEM 4000 blood gas analyzer is simple to use, suitable to be managed by non-expert users of analytical systems such as professional nurses and doctors, has a high degree of automation, internal calibration and quality control systems; it is free of maintenance, safe for operators and able to interface with the laboratory computer system (LIS) and with data management computer systems. The analyzers used in our study include a new generation, completely autonomous quality control system called iQM (intelligent Quality Management) capable of controlling all phases of the analytical process at any moment. The iQM also manages the possible presence of non-conformities of the sample, the cartridge, the software and activates automated corrective actions by tracing all the operations performed; any out-of-control parameters are temporarily disabled until the problem is resolved.

### 2.3. Non-Conformities (NC) of the Pre-Analytical Phase

The following non-conformity reports have been identified: (a) “**sample detection**” error due to inadequate mixing of the sample before carrying out the analysis (type 1 error); (b) “**insufficient sample**” due to the use of samples with an unsuitable volume (type 2 error); (c) “**sample not detected**” error due to incorrect insertion of the aspiration probe inside the syringe or error due to failure to expel air bubbles (type 3 error); (d) “**microcoagulum detection in the sample**” error due to probable incorrect anticoagulation of the sample also caused by inadequate mixing of the sample immediately after collection (type 4 error). These NC were selected for statistical evaluation as they are related to non-compliance with the operator’s EGA sample management procedures. Alarms not dependent on the operator, such as the presence of interferers or errors in the calibration and quality control phase, have been excluded.

### 2.4. Supervision and Centralized Management of Blood Gas Analyzers

All blood gas analyzers located in the departments of the Bari Polyclinic, including those in the intensive care departments, are connected to the company intranet and interfaced with the GEMweb Plus (Werfen Instrumentation Laboratory, S.p.A, Milan, Italy) data management system, which is in turn connected to the laboratory LIS for the acquisition, transmission and traceability of demographic data, patient results and POC data. The GEMweb Plus system allows the Core Laboratory to remotely supervise all the decentralized POC stations connected to the network in real time; constantly check the functional status of the individual instruments and of the panel of enabled analytes; verify the analytical performance of each analyzer by recalling reports about the quality controls performed; view the blood gas analyses performed in the different departments, the patient results and the history for each individual patient. Furthermore, the Core Laboratory can recall the reports on the count of the samples processed by the single analyzer in the time interval considered, view and monitor process non-conformities, the related corrective actions taken and their outcome.

### 2.5. Report Management

The Core Laboratory coordinators periodically briefed the resuscitation staff on the performance of each analyzer. It was essential to show the value of the activity in a specific period of time by disclosing information on the operation.

### 2.6. Statistic Analysis

Categorical variables were expressed in terms of absolute (mean, median, SD and Confidence Interval (CI)) and relative (percentiles and percentage changes) proportions. The **Kolmogorov–Smirnov test** was used to assess the normality of distribution of investigated parameters (number of NC for each grading, reported in the initial and second period). **Mann–Whitney test** was used to analyze any significant differences in terms of media of NC encountered in the pre-COVID vs. COVID period. *p* values < 0.05 were considered statistically significant.

The **Chi-square test** was used to test hypotheses about frequency difference; *p* values < 0.05 were considered statistically significant.

The comparison between the data in the pre-COVID and COVID periods was graphically illustrated with a multiple line graph. **Box and Whisker plot** was used for graphical comparison of data. Statistical analysis was performed using Medcalc (Software Inc., California, USA).

### 2.7. Performance Evaluation

At the end of each period, the data on the NCs of the blood gas analyzers were processed and analyzed to identify the performance limits for the following period. Statistical data on the value of the NC relating to the specific periods identified were used for the calculation of the Six-Sigma Metric for performance classification; the result was evaluated according to the specifications reported in the literature. Typical performance of acceptability of the reporting error of the sample was taken from the literature [17] and from consultation of the site https://www.westgard.com/six-sigma-calculators.htm (accessed on 9 December 2022).

## 3. Results

In the resuscitation units in the pre-COVID period, 16,632 blood gas tests were performed on the two analyzers supplied; in the COVID period, 31,364 blood gas tests were performed on the three analyzers supplied. Particular attention was paid to the pre-analytical phase in relation to the ability of the iQM system integrated in the GEM 4000 blood gas analyzers to detect non-conformity conditions (NC). The non-conformities related to the suitability of the EGA sample and management by the operators scored 652 (3.9%) and 749 (2.4%) in the pre-COVID and COVID periods, respectively. All data were stratified by a one-month collection period. Table 1 shows the summary statistical value, reporting the range (minimum and maximum), the mean, the median, the calculation of the 95% CI and the 25–75th percentile and normal distribution (Kolmogorov–Smirnov test). The “sample detection” error (type 1 error) was equal to n. 418 (pre-COVID) and n. 517 (COVID); the “insufficient sample” (type 2 error) was equal to n. 104 (pre-COVID) and n. 106 (COVID); the “sample not detected” error (type 3 error) was equal to n. 65 (pre-COVID) and n. 83 (COVID); the “microcoagulum detection in the sample” error (type 4 error) was equal to n. 65 (pre-COVID) and n. 43 (COVID). The statistical evaluation of the number of non-conformities obtained in all subjects and in the two groups stratified according to the pre- and pandemic period is shown in Table 1. Across all results, the lowest and highest number of NC for single registration were, respectively, no. 0 (type 3 error of the pre-COVID period; type 4 pre-COVID and type 4 COVID) and no. 57 (type 1 error of the COVID period). The Kolmogorov–Smirnov test showed a normal distribution of total NCs reported in the COVID period (*p* = 0.0005) vs. total NC reported in the pre-COVID period.

The Box and Whisker plot, used to visualize the statistical summary of the distribution of the individual NC classes and for the graphical comparison of the data, is shown in Figure 1. The multiple line graph reports the consecutive observations of the total NC differentiated by serial observation periods. The moving average trend line obtained as the average of three consecutive data points is shown in the graph and shows a stable trend for data collection and a peak increase in the January–April period of both periods evaluated. This may be related to increased hospitalizations for complicated seasonal viral infections (pre-COVID period) and to the 2020 pandemic shown in Figure 2. Table 2 shows the results of the Mann–Whitney U test for the comparison between the means of the different differentiated NC per observation period. The evaluation showed a statistically significant difference (*p* < 0.05) between NC with type 4 error recorded in the pre-COVID period versus the COVID period.

The hypothesis testing on the frequency difference was carried out using the Chi-squared test with the following values: Chi-squared 10.507; DF 3; Contingency coefficient 0.0863; Significance level *p* = 0.0147. The calculated *p*-value is small (<0.05), showing a significant relationship between the various NC groups.

### Performance Evaluation

The calculation of the 25th and 75th percentiles used as performance limits calculated at the end of each data collection period provided results of 23.50 to 30.50 for the pre-COVID period vs. 22.00 to 35.50 of the NCs for the COVID period. Between these two periods, non-overlapping of the distribution of the absolute number of recorded events was observed.

The calculation of the Six-Sigma Metric for performance classification provided defects per million of 39,202 and 23,881 and Sigma-Metric of 3.3 and 3.5 for the pre-COVID period vs. the COVID period, respectively. The data obtained, compared with the qualitative objective calculated for analytical performance set between 3 and 4, were considered satisfactory and improving [17].

## 4. Discussion

Point of Care (POC) analysis refers to clinical laboratory tests that are typically performed outside of a clinical laboratory, closer to the patient and sometimes at the patient’s bedside. This includes all tests performed by non-laboratory personnel, such as nurses, physicians and respiratory therapists, regardless of the testing site. Point of Care analysis is widely used in hospital settings, even in clinical scenarios where a fast turnaround time is needed for results or if central laboratory testing is not available [18,19]. Recent documents provide guidance to both the hospital and POC users on the activities necessary to obtain quality, accurate, precise and clinically valid POC results. Aspects of quality assurance include analytical quality management, such as internal quality control (IQC) and external quality assurance (EQA) and all management system documents [20,21]. In compliance with the regulatory and accreditation requirements in force and with the recommendations of the guidelines produced in the POC field, the laboratory of the Clinical Pathology Unit of the Hospital Policlinic of Bari, Italy (Core Laboratory), has developed a quality assurance program based on analytical quality controls by IQC and EQA, the development of quality management documents and clinical risk monitoring. To achieve the following objectives, it has used sophisticated technology and solutions that have allowed optimal quality management of the complete diagnostic process and of the results generated by the blood gas analysis activities performed in the departments [21,22].

Analytical quality in the POC device field is a fundamental prerequisite for the reliability of the results, [18]. However, the data collected in recent years have shown the need to re-evaluate in particular the pre-analytical phase, a key element of the total test process (TTP) [21,22]. All experts agree on non-analytical performance monitoring through the collection and analysis of specific and valuable quality indicators (QI) [22,23]. While analytical Quality Indicators (QI) based on IQC and EQA performance criteria are standardized and easily applied, the development and use of reliable indicators for the extra-analytical phase is still in its infancy. Core laboratories should identify critical activities and implement QI in order to monitor and achieve improvements [22,23]. The ultimate goal is to keep the risk of error to a minimum level (8–21). Our study was carried out during the health emergency of the COVID-19 pandemic when an increase in hospitalizations at the Hospital Policlinic of Bari, Italy, was registered. A progressive increase both in the number of beds dedicated to critically positive SARS-CoV-2 patients and in the turnover of medical staff did not influence the quality of the Total Testing Process (TTP) including the pre-analytical phase of blood gas analysis.

The type and incidence of pre-analytical operator-operated NC that occurred in the resuscitation departments of the Polyclinic from the beginning of the pandemic (March 2020) until February 2022 did not show statistically significant differences in terms of average and percentage differences when compared with those recorded in the pre-COVID period (March 2018–February 2020).

Despite the facts that POC devices and Core Laboratory testing are different and POC users are non-laboratory professionals, it was possible to perform an accurate evaluation.

In fact, it is the competence of the users that creates new challenges for the achievement of specific quality objectives. The frequency, distribution and impact of patient safety risks in the POC device TTP are different than in the Core Laboratory; however, proper device management and operator expertise are effective tools to overcome this challenge and improve quality [1,24]. Our study highlighted four types of NC in the pre-analytical phase related to operator procedures and non-compliance with the indications contained in the operating instructions. The data were measurable as they were available in the data management system. Then, they could be used for statistical evaluations; therefore, the QI of the blood gas analysis process was useful.

The number of NC obtained from our evaluation in terms of the large number of total events and presented as a percentage of defects was low, but needed confirmation [25]. Therefore, the NC were also measured as defects per million and expressed on the Six Sigma scale (0–6). Six Sigma, initially developed in industry to evaluate product quality, measures the degree of deviation from the goal in any process [25]. A value of 3-Sigma performance is considered the minimum for any industrial process. The first goal of a Six Sigma project in business and industry is usually to improve Sigma [17].

The Six Sigma scale has been recognized as a useful tool for evaluating laboratory quality management systems [21,26] and can be applied to virtually any pre-analytical, analytical or post-analytical process. The observed error rates for many of these processes are in the 3 to 4 Sigma range. When QIs are presented on the Six Sigma scale, they are simple to accept and directly related to quality level and requirement satisfaction [25]. The IQ analyzed in our data collection by Sigma evaluation did not show an increase in error rates despite the complexity of the processes initiated in the management of patients with SARS-CoV-2 infection and the increase in the number of tests performed. Furthermore, this criterion, based on state of the art of quality indicators (QI) and on the indications provided by “a consensus statement on behalf of the IFCC Working Group”, allows for aligning the performance specifications with the general improvement path of the POC device activities.

A consequent consideration is that in our experience the risks have been faced and limited by using effective training procedures in maintaining the skills of the operators even with modified work turnover. During the period of the COVID-19 pandemic, clinical users of the POC analyzer for EGA complied with the indications and procedures shared with the central laboratory without increasing the operator-dependent NC. The Central Laboratory has developed and documented monitoring activities capable of guaranteeing the maintenance of quality at suitable standards and the development of improvement strategies in the event of adverse events.

## 5. Conclusions

This study showed a systematic approach to identify and monitor POC device measurement IQs for EGA. The evaluation aimed to verify if, during the COVID-19 pandemic, with the increase in EGA requests, there had been a significant increase in the error rate committed by staff.

Repeated testing due to errors logged, many times a day, coupled with pure practice of EGA testing also contributed to the significant overall drop in staff error rate.

Regular monitoring of the quality indicators has been found to be an appropriate measure of the TTP quality of the POC analyzer for EGA.

## Figures and Tables

**Figure 1 diagnostics-13-01044-f001:**
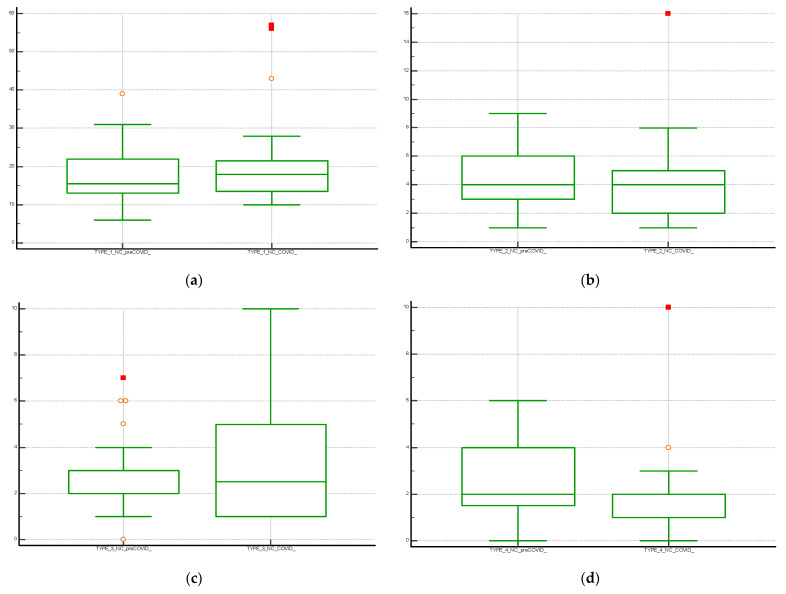
Boxplot with statistical summary obtained from the distribution of the individual classes of non-conformities (NC); (**a**) NC Type 1; (**b**) NC Type 2; (**c**) NC Type 3 and (**d**) NC Type 4. The central box shows the values from the 25th to the 75th quartile, the central line the median, the horizontal lines the extension from the value.

**Figure 2 diagnostics-13-01044-f002:**
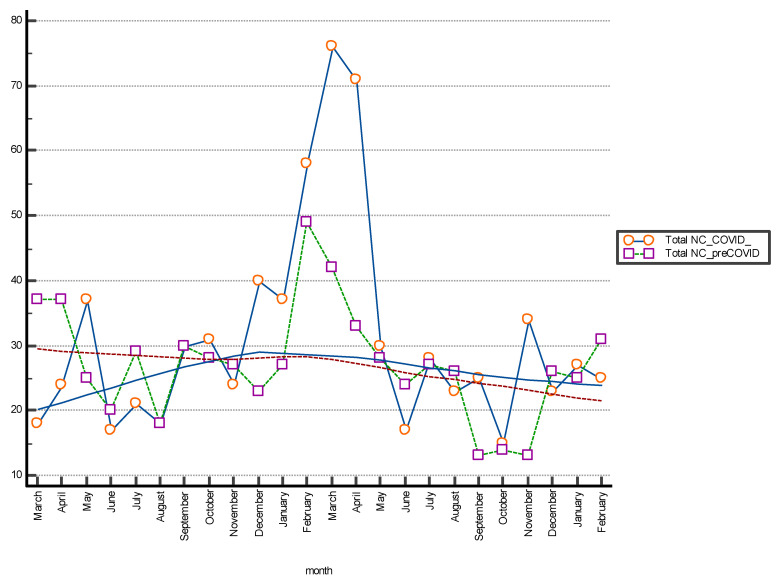
The Multiple line graph shows the number of total non-conformities (NC) differentiated by month of observation; period March 2018–February 2022. The moving average trend lines are indicated.

**Table 1 diagnostics-13-01044-t001:** Summary statistical value reporting the range (minimum and maximum), the mean, the median with calculation of the 95% CI; the 25–75th percentile of the normal distribution (Kolmogorov–Smirnov test). The non-conformities recorded in the evaluation period March 2018–February 2020 are indicated with NC pre-COVID and the non-conformities recorded in the period March 2020–February 2022 with NC_COVID.

	N Months	Minimum	Maximum	Mean	95% CI	Median	95% CI	25–75 P	Normal Distribution
**Total NC-pre-COVID**	24	13	49	27.17	23.53 to 30.79	27	24.74 to 29.25	23.50 to 30.50	0.3133
**Total NC-COVID**	24	15	76	31.21	24.43 to 37.98	26	23.00 to 31.76	22.00 to 35.50	0.0005
**TYPE1 NC-pre-COVID**	24	6	39	17.42	14.22 to 20.61	15.5	14.00 to 22.00	13.00 to 22.00	0.025
**TYPE1 NC-COVID**	24	10	57	21.54	16.11 to 26.96	18	15.49 to 20.25	13.50 to 21.50	0.0001
**TYPE2 NC-pre-COVID**	24	1	9	4.33	3.46 to 5.20	4	3.00 to 5.25	3.00 to 5.00	0.2711
**TYPE2 NC-COVID**	24	1	16	4.42	3.11 to 5.71	4	2.74 to 5.00	2.00 to 5.00	<0.0001
**TYPE3 NC-pre-COVID**	24	0	7	2.71	1.96 to 3.45	2	2.00 to 3.00	2.00 to 3.00	0.0603
**TYPE3 NC-COVID**	24	1	10	3.46	2.31 to 4,59	2.5	1.74 to 4.25	1.00 to 5.00	0.088
**TYPE4 NC-pre-COVID**	24	0	6	2.71	1.93 to 3.48	2	2.00 to 4.00	1.50 to 4.00	0.4568
**TYPE4 NC-COVID**	24	0	10	1.79	0.92 to 2.65	1	1.00 to 2.00	1.00 to 2.00	<0.0001

**Table 2 diagnostics-13-01044-t002:** Comparison of the difference between the means of total NC or differentiated according to the type encountered in the pre-COVID vs. COVID periods (Mann–Whitney test).

	Total NC	TYPE 1 NC	TYPE 2 NC	TYPE 3 NC	TYPE 4 NC
NC-pre-COVID vs. NC-COVID	*p* = 0.8364	*p* = 0.3017	*p* = 0.6991	*p* = 0.5835	*p* = 0.0278

## Data Availability

All data, stored anonymously, are available to all those who explicitly request it from the corresponding author.

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
