# Peer review of "Analysis of Quality Indicators of the Pre-Analytical Phase on Blood Gas Analyzers, Point-Of-Care Analyzer in the Period of the COVID-19 Pandemic"

_diagnostics, 2023, doi:10.3390/diagnostics13061044_

Round 1

Reviewer 1 Report

The authors evaluated and compared blood gas analysis (EGA) non-conformities (NC) considered operator dependent as quality indicators (IQ) of the pre-analytical phase. The study design is simple. Although the conclusions are predictable, they still may be useful for the clinician. However, the authors misunderstand the point-of-care testing (POCT) and the study doesn’t displace any relationship with POCT. Therefore, a major revision is necessary before its publication in Diagnostics.

1.     The main concern is what is the link between this study with POCT? Not all decentralized tests are POCT. At first, the used method or device must be in accord with the “ASSURED” principle. However, I don’t think the used GEM 4000 series blood gas analyzer is a POCT device. There, the title and the related descriptions in the manuscript should revised.

2.     In table 2, please use the correct mathematical symbol, where “,” should be “.”.

3.     The language in this manuscript should be carefully polished and there are a lot of sentences with grammatical problems. For instance, “The evaluation aimed to verify whether the period of the COVID 19 pandemic which led to substantial organizational and management changes at the Hospital Polyclinic of Bari Italy could result in an additional risk factor capable of highlighting inadequacies in the quality management program launched by the Core Laboratory” in Conclusions.

4.      This design of study is easy-to-understand, while the obtained conclusions are described vaguely. Please simply and clear list the conclusions in the manuscript.

Author Response

dear editor

In replay review 1 to "The main concern is what is the link between this study with POCT? Not all decentralized tests are POCT. At first, the used method or device must be in accord with the “ASSURED” principle. However, I don’t think the used GEM 4000 series blood gas analyzer is a POCT device. There, the title and the related descriptions in the manuscript should revised." Thanks for the reviewer's observation, we have corrected the text.

In replay to "In table 2, please use the correct mathematical symbol, where “,” should be “.”. We have changed the text by inserting the period instead of the comma.

In replay to "The language in this manuscript should be carefully polished and there are a lot of sentences with grammatical problems. For instance, “The evaluation aimed to verify whether the period of the COVID 19 pandemic which led to substantial organizational and management changes at the Hospital Polyclinic of Bari Italy could result in an additional risk factor capable of highlighting inadequacies in the quality management program launched by the Core Laboratory” in Conclusions. We have checked all the text for any grammatical errors. We have also improved, also in agreement with reviewer 2, the paragraph "conclusions" making them more schematic

In repaly to "This design of study is easy-to-understand, while the obtained conclusions are described vaguely. Please simply and clear list the conclusions in the manuscript." We have modified the text following the indications of reviewer 1

Reviewer 2 Report

The manuscript consists of total 10 pages, including 2 figures, 2 tables and the list of total 25 literature references. The article presents the original results of the study on the non-conformity depending on the operator that were recorded by the medical laboratory equipment utilized for the point-of-care blood gas testing both during the COVID-19 pandemics and before. As such, it is both current and interesting for the Readers and fits into the scope of works published by the Journal. The title of the manuscript is clear and relevant to the contents of the article. The English language quality is acceptable, the structure of the text is logical.

The Introduction section is rather concise but informative enough, provides the needed core information on the project background and justification.

The Materials and methods section is structured

The Results are consistent with the declared methodology, clearly presented, supported adequately with tables and figures.

The Discussion section places the own results into the context of previously published knowledge.

The Conclusions are concise and clear enough. However, it would be worth it to add that in the COVID-19 pandemic time, in comparison to the earlier period, the blood gas testing was not only intensely trained, supervised and managed, which no doubt is true, but also it became a procedure that was very often carried out by the staff. It is always good to assume that the quality management and improvement strategies that were introduced have worked - however in my opinion, based on the real-life observation, during the COVID-19 pandemic the on-site blood gas testing became all of the sudden even more than before a mundane and burdensome everyday routine but it was executed under the increased pressure to avoid repeating the failed tests: the repeated and thus polished many times a day by sheer practice blood gas testing procedures together with the relentless feedback from the machines in case of any detected errors and the resulting merciless demand to repeat the failed tests that consumed the precious resources under the pressure of always insufficient time and stress, they all together resulted in the observed overall significant drop of the error rate committed by the staff.

The tables and figures adequately present the key data and thus add to the overall clarity of the article, and their captions are informative enough.

The literature references are relevant to the topic of the article and reasonably recent.

The Authors might want to consider including into the article the remarks on the basis of the importance of on-site blood gas testing availability in COVID-19 pandemic, as in: https://doi.org/10.3390/covid2010004  https://doi.org/10.3390/biology10090852 https://doi.org/10.3390/jcm10122731

Author Response

dear editor

In replay to review 2  : "we are glad to have received an important consensus  from reviewer 2. This rewards us for  the enormus efforts required to realize this  manuscript.

In repaly to "The Authors might want to consider including into the article the remarks on the basis of the importance of on-site blood gas testing availability in COVID-19 pandemic, as in: https://doi.org/10.3390/covid2010004  https://doi.org/10.3390/biology10090852 https://doi.org/10.3390/jcm10122731" The reference items suggested by reviewer 2 have been inserted into the text

Round 2

Reviewer 1 Report

No comments any more.